# Prediction of graft loss in living donor liver transplantation during the early postoperative period

Raiki Yoshimura[1☯], Naotoshi Nakamura[1,2☯], Takeru Matsuura[1☯], Takeo Toshima[3], Takasuke Fukuhara[4], Kazuyuki Aihara[5], Katsuhito Fujiu[6,7], Shingo Iwami[1,5,8,9,10,11,12☯]*, Tomoharu Yoshizumi[3☯]*

1 Interdisciplinary Biology Laboratory (iBLab), Division of Natural Science, Graduate School of Science, Nagoya University, Nagoya, Japan, 2 Department of Data Science, Yokohama City University, Yokohama, Japan, 3 Department of Surgery and Science, Graduate School of Medical Sciences, Kyushu University, Fukuoka, Japan, 4 Department of Virology, Faculty of Medical Sciences, Kyushu University, Fukuoka, Japan, 5 International Research Center for Neurointelligence, The University of Tokyo Institutes for Advanced Study, The University of Tokyo, Tokyo, Japan, 6 Department of Cardiovascular Medicine, Graduate School of Medicine, The University of Tokyo, Tokyo, Japan, 7 Department of Integrative Physiology, Institute of Science Tokyo, Tokyo, Japan, 8 Institute of Mathematics for Industry, Kyushu University, Fukuoka, Japan, 9 Institute for the Advanced Study of Human Biology (ASHBi), Kyoto University, Kyoto, Japan, 10 Interdisciplinary Theoretical and Mathematical Sciences Program (iTHEMS), RIKEN, Saitama, Japan, 11 NEXT-Ganken Program, Japanese Foundation for Cancer Research (JFCR), Tokyo, Japan, 12 Science Groove Inc., Fukuoka, Japan

☯ These authors are contributed equally to this study.

* iwami.iblab@bio.nagoya-u.ac.jp (SI); yoshizumi.tomoharu.717@m.kyushu-u.ac.jp (TY)

## Abstract

Liver transplantation is almost the only way to save patients with end-stage liver disease. Particularly, living donor liver transplantation (LDLT) has gained importance in recent years thanks to the shorter waiting times and better graft quality than with deceased donor liver transplantation (DDLT). However, some patients experience graft loss due to unexpected infections, sepsis, or immune-mediated rejection of the transplanted organ. An urgent need exists to clarify which patients experience graft loss. Several models have been proposed, but most analyze the classic DDLT, and knowledge about LDLT is lacking. In this study, we retrospectively analyzed clinical data from 748 patients who underwent LDLT. By adapting machine learning methods, we predicted early graft loss (within 180 days postoperatively) with better performance than conventional models. The model enabled us to stratify a highly heterogeneous sample of patients into five groups. By focusing on survival time, we next categorized the patients into three groups with early, intermediate, and late or no graft loss. Notably, we identified the intermediate-loss group as a distinct population similar to the early-loss population but with different survival times. Additionally, by proposing a hierarchical prediction method, we developed an approach to distinguish these populations using data up to 30 days postoperatively. Our findings will enable the early identification of individuals at risk of graft loss, particularly those in

**Data availability statement:** The data supporting the study findings are available from the Section of IRB & Ethics Committee Administration, Kyushu University (byskenkyu@jimu.kyushu-u.ac.jp) upon reasonable request. However, partial restrictions apply. Some data contain sensitive patient information and are subject to confidentiality agreements, which may limit the extent of data sharing. The codes used for analyses are available via following github url: https://github.com/ta-mats/MT-LDLT/tree/main.

**Funding:** This study was supported in part by Scientific Research (KAKENHI) B JP23K28187 (to S.I.); Grant-in-Aid for Challenging Research (Exploratory) JP24K22338 (to S.I.); Japan Agency for Medical Research and Development (AMED) JP24gm1310002, JP24fk0108684, JP24fk0108685, JP24fk0410052, JP24fk0310504, JP24wm0425011, JP24ek0109707, JP24wm0625302 (to S.I.); JST MIRAI JPMJMI22G1 (to S.I.); Moonshot R&D JPMJMS2021 (to K.A. and S.I.) and JPMJMS2025 (to S.I.); Institute of AI and Beyond at the University of Tokyo (to K.A.); SECOM Science and Technology Foundation (to S.I.). The funders had no role in study design, data collection and analysis, decision to publish, or preparation of the manuscript.

**Competing interests:** The authors have declared that no competing interests exist.

the early- and intermediate-loss groups. This will allow for appropriate patient care, such as switching to DDLT, identifying other living donors for LDLT, or preparing for re-transplantation, leading to a bottom-up improvement in transplant success rates.

## Author summary

Liver transplantation is crucial for treating end-stage liver disease, with living donor liver transplantation (LDLT) offering advantages like shorter waiting times and better graft quality compared to deceased donor liver transplantation (DDLT). However, graft loss remains a concern, often due to post-LDLT infection or immune-mediated rejection. To address this, we analyzed 748 LDLT cases using machine learning to predict early graft loss (within 180 days) with greater accuracy than conventional methods. Additionally, patients were stratified into five groups and further categorized into three survival-based groups with early, intermediate, or late/no graft loss. The intermediate-loss group shared characteristics with the early-loss population but had longer survival times. This hierarchical approach enables early risk identification, thereby supporting clinical decisions, including DDLT registration or re-transplant evaluation for patients at risk for early graft loss and intensified monitoring for patients at risk for intermediate graft loss.

## Introduction

The liver plays such a vital role in functions like nutrient and toxin metabolism, blood volume regulation, and immune system support that liver dysfunction, whether from hepatocellular carcinoma (HCC), metabolic-associated steatosis liver disease, or other causes, can lead directly to death [1]. Consequently, liver transplantation is a critical treatment option and often the only way to save patients with end-stage liver disease. Until now, deceased donor liver transplantation (DDLT) has been the predominant approach, especially in Western countries. Yet, the significance of living donor liver transplantation (LDLT) has risen in recent years thanks to shorter waiting times, better graft quality, and improved post-transplant outcomes compared with DDLT [2]. As a result, LDLT has increasingly been adopted not only in Asia, where the supply of deceased donor livers is limited, but also in the United States and Europe. Nevertheless, LDLT still presents unresolved challenges. Unlike DDLT, LDLT involves transplantation of partial liver grafts, which introduces unique surgical and physiological challenges. For example, small-for-size syndrome caused by limited graft volume and hemodynamic imbalance can lead to elevated portal pressure, hepatocellular injury, and early graft loss [3]. Additionally, LDLT is associated with a higher incidence of biliary complications due to anatomical variations such as multiple bile duct orifices and increased technical difficulty in anastomosis [4,5]. These risks differ substantially from those in DDLT, highlighting the need for predictive models specifically tailored to the clinical context of LDLT.

Early graft loss and post-transplant mortality have been strongly associated with early allograft dysfunction (EAD) [6], but it remains unclear which individuals are more susceptible to graft loss. Various predictive models, such as the D-MELD score [7], have been developed, but most of them focus on DDLT. Although several models specific to LDLT, including predictive scores [8,9], TB-INR criteria using the definition of EAD [10], and the eGLR score [11], have also been proposed, they are generally based on simple univariate or multivariate analyses and may not fully capture the complex interplay of factors contributing to early graft loss or the significant heterogeneity among patients. Moreover, most studies focused on statistically comparing patients with early graft loss with other patients, leaving the generalizability of these models to validation data uncertain.

In our study, we conducted a quantitative analysis of extensive clinical data from both recipients and donors in a cohort of 748 LDLT patients at Kyushu University Hospital. Data were collected preoperatively, intraoperatively, and up to 30 days postoperatively. We specifically focused on early graft loss and developed a highly accurate machine learning model by dividing the patients into derivation and validation cohorts. Through this model, we identified five patient groups with low heterogeneity, despite significant variation among individuals. By further refining these into three groups based on graft survival time, we developed a machine learning model that categorizes patients into groups with early, intermediate, and late or no graft loss. Our stratification and prediction model will enhance optimal resource allocation and personalized medicine, offering new insights into the mechanisms of graft loss in LDLT.

## Results

### Description of cohort and study design

We used extensive clinical data collected from 748 patients who underwent LDLT at Kyushu University Hospital between 1997 and 2023. This hospital receives patients requiring LDLT from all over western Japan, including the large areas of Kyushu, Okinawa, and the western Chugoku region, home to 15 million people. More than 1,000 LDLTs have been performed, representing 10% of all LDLTs in Japan, and the patient population is widely heterogeneous in terms of severity and etiology.

The clinical data, which were annotated so that we could assess whether the graft was lost or not, included preoperative, intraoperative, and postoperative factors for both donors and recipients. In particular, the postoperative data included data up to 30 days after LDLT only to enable early prediction of transplanted graft loss. Each data set included basic demographic information (such as age and sex), blood test results, and other information (see **Tables A**, **B,** and **C in** S1 Text for details).

We defined "early graft loss" as graft loss within 180 days after LDLT. The analysis of graft survival time for all patients experiencing graft loss (n = 177) showed that about 22% (n = 39) of patients experienced early graft loss (see **Fig 1A**).

To assess the validity of our developed model, we randomly divided the cohort into a derivation cohort (n = 563) and a validation cohort (n = 185) (see **Fig 1B**). Note that there were no significant differences between the derivation and validation cohorts for most variables (see **Table A in** S1 Text). Further details on data preparation are outlined in the Methods.

### Prediction of individual-level early graft loss in living donor liver transplantation

Early graft loss is primarily due to EAD and is thus an indicator of graft fixation in liver transplantation [6]. We first attempted to predict early graft loss, as defined above, at the individual level using all variables from donors and recipients (i.e., preoperative, intraoperative, and postoperative data as listed in **Tables A**, **B,** and **C in** S1 Text) for patients experiencing LDLT. We trained four classification models, specifically logistic regression, Random Forest (RF), lightGBM, and XGBoost, using the derivation cohort data and evaluated the predictive performance of the model using the validation cohort data (see **Fig 1B** and Methods). These four models were consistently evaluated in subsequent analyses. Among these models, RF achieved the highest predictive performance, with an area under the receiver operating characteristic

**Fig 1. Prediction of early graft loss at the individual level and identification of crucial factors: (A) Survival curves for all patients who experienced graft loss are plotted.** The shaded area represents the 95% confidence interval of the curve. The black line marks 180 days post-LDLT, indicating the threshold for early graft loss. **(B)** A workflow for the development and evaluation of individual-level prediction models is shown. All data

were randomly split into derivation (75%) and validation (25%) cohorts, respectively. For each graft loss prediction, binary classifiers was constructed and trained to predict early graft loss by using derivation cohort data and subsequently the performance of the model was evaluated by using validation cohort data. This figure was created by hand. **(C)** The ROC curve for a RF classifier trained to predict early graft loss (within 180 days post-LDLT) is shown alongside previously proposed models for comparison. The corresponding ROC-AUC value of a RF classifier is shown at the top of the panel. **(D)** Feature importance of the predictive model in C is illustrated as a SHAP summary plot. The y-axis represents the clinical items, arranged in order of their contribution to the prediction. The contribution of each feature for each patient (each point) to the prediction is represented as SHAP values (x-axis). A higher SHAP value means a higher contribution to the likelihood of early graft loss, while a lower value indicates a higher contribution to late or no graft loss. The color of each point in each feature represents the value of that feature for the patient, with higher values shown in red and lower values in blue. Abbreviations and definitions: 30POD, 30 postoperative days; 14POD, 14 postoperative days; ascites, ascites volume; T-BIL, serum total bilirubin; PT%, prothrombin time activity percentage; D max T-BIL, donor's serum total bilirubin; splenectomy, the spleen was removed during surgery. **(E)** The ROC curve of a RF classifier trained to predict early graft loss using only the six crucial factors extracted using SHAP values. The corresponding ROC-AUC is calculated and displayed at the top of the panel. **(F)** The preoperative, intraoperative, and postoperative data selected as the six crucial factors by SHAP for early graft loss shown in E are plotted and colored accordingly. Blue dotted lines indicate the normal reference ranges for each variable: ascites (30POD): 0, ascites (14POD): 0, T-BIL(14POD): 0.2-1.3, PT%(14POD): 70-130, and D max T-BIL: 0.2-1.2. Statistical significance is calculated using the Wilcoxon rank-sum test for continuous values and p-values are expressed as follows (N.S.: p-value > 0.05, *: p-value ≤ 0.05, **: p-value ≤ 0.01, and ***: p-value ≤ 0.001, respectively). Also, Fisher's exact test for categorical values is used and whether there were significant differences in the proportions of each group relative to the total was examined (N.S.: p-value > 0.05, *: p-value ≤ 0.05, **: p-value ≤ 0.01, and ***: p-value ≤ 0.001, respectively). "TRUE" indicates that the item is applicable, e.g., in the context of splenectomy, TRUE means cases with splenectomy at the time of LDLT, FALSE means cases without splenectomy at the time of LDLT. Orange represents the percentage of TRUE and gray represents the percentage of FALSE. **(G)** The ROC curves of RF classifiers trained to predict long-term graft loss by using all variables (black curve) or the six important factors (blue curve) for early graft loss prediction are plotted and colored, respectively. The corresponding ROC-AUC is calculated and displayed on the legend in the panel.

curve (ROC-AUC) of 0.90 (**Table 1**). Furthermore, when compared with conventional prediction models such as the predictive score, D-MELD, TB-INR, and eGLR score, RF clearly outperformed these approaches, demonstrating substantially higher discriminative ability (see **Fig 1C** and **Table 2**). This result remained robust across different random seeds, as confirmed by sensitivity analyses and cross-validation within the derivation cohort data (see **Fig A in** S1 Text).

To further assess clinical utility, we examined calibration and performed decision curve analysis (DCA) for early graft loss prediction (**Fig B in** S1 Text). The calibration plot demonstrated generally good agreement between predicted and observed probabilities, although slight underestimation was observed overall. Notably, the curve dropped at the higher end of predicted probabilities due to a limited number of high-risk cases. The DCA revealed a positive net benefit when

**Table 1. Comparison of performance of four machine learning models for early graft loss prediction.**

| Method | Accuracy | Precision | Recall | F1 score | ROC-AUC |
|---|---|---|---|---|---|
| RF | 0.79 | 0.59 | 0.89 | 0.60 | 0.90 |
| Logistic Regression | 0.71 | 0.55 | 0.74 | 0.52 | 0.76 |
| XGBoost | 0.67 | 0.56 | 0.83 | 0.51 | 0.87 |
| Lightgbm | 0.83 | 0.60 | 0.86 | 0.62 | 0.89 |

**Table 2. Comparison of prediction performance with previously reported metrics.**

| Method | Accuracy | Precision | Recall | F1 score | ROC-AUC | DeLong's† |
|---|---|---|---|---|---|---|
| RF | 0.82 | 0.59 | 0.89 | 0.60 | 0.90 | – |
| Predictive score | 0.88 | 0.54 | 0.55 | 0.54 | 0.56 | p < 0.001 |
| D-meld | 0.93 | 0.47 | 0.49 | 0.48 | 0.51 | p < 0.001 |
| TB-INR criteria | 0.95 | 0.48 | 0.50 | 0.49 | – | – |
| eGLR score | 0.65 | 0.49 | 0.46 | 0.43 | 0.55 | p < 0.001 |

† : DeLong's test was used to compare the ROC-AUC of the proposed model (Random Forest) with those of existing models. The TB-INR criteria were excluded from comparison as an ROC curve could not be constructed.

using thresholds in the range of 0.10 to 0.15, suggesting that the model may support clinically meaningful decision-making in early postoperative management.

Collectively, these results indicate that the data up to 30 days postoperatively contained sufficient information to predict early graft loss and that machine-learning-based methods can capture complex signs of early graft loss more effectively than univariate or multivariate analyses.

We then computed SHapley Additive exPlanations (SHAP) values to evaluate the significance of clinical data in discerning between patients with early graft loss and all other patients (i.e., "others"). This computation was based on the RF classifier for early graft loss (Fig 1C). By using SHAP values to narrow down the number of variables needed (see Methods), we found that the model trained using the information for only the six variables listed in Fig 1D could predict graft loss as accurately as the model trained using all variables (Fig 1E; ROC-AUC of 0.90). Interestingly, even when donor information was excluded, model performance was still good with an ROC-AUC of 0.88 (see **Fig C in** S1 Text). In this setting, the set of important predictors was revised to include seven variables—consisting of the top five factors excluding donor information, along with two additional non-donor-related factors (see **Fig C in** S1 Text). These results indicate that the indicators of early graft loss are concentrated in the information derived from these six features, or the seven features of the recipients only.

Regarding the six features shown in Fig 1D, we compared each value between patients with early graft loss and the others as depicted in Fig 1F. We found that early graft loss was associated with poorer performance status before LDLT, increased ascites, lower prothrombin time activity percentage (PT%), and higher serum total bilirubin levels (T-BIL) after LDLT (see Fig 1F). Ascites, reduced PT%, and elevated T-BIL are typical symptoms of small-for-size syndrome following LDLT. Interestingly, donor and graft information did not significantly contribute to the prediction of early graft loss (**Fig C in** S1 Text), suggesting that the onset of small-for-size syndrome and related graft loss may depend more on the recipient's condition than on the donor's (see Discussion).

### Long-term prediction of individual-level graft loss in living donor liver transplantation

We found that early graft loss can be predicted at the individual level using six crucial factors, including data up to 30 days after LDLT. Excluding patients with early graft loss (n = 30) from the derivation cohort (n = 563), we then attempted to predict long-term graft loss in the remaining 533 patients whose grafts survived more than 180 days after LDLT, thus distinguishing between long-term graft loss and graft survival.

We first applied RF using the six crucial factors identified in early graft loss prediction to predict long-term graft loss as well. However, the performance of this model was relatively low, with an ROC-AUC of 0.52 (blue curve in Fig 1G). This result suggests that different factors contribute to graft loss 180 days after LDLT. We then applied the four classifiers (i.e., logistic regression, RF, LightGBM, and XGBoost) using all variables as well. Among them, RF achieved the highest performance, with an ROC-AUC of 0.78 (black curve in Fig 1G and Table 3), although the improvement was only marginal.

Since using all variables did not significantly enhance predictive performance, follow-up of longer than 30 days after LDLT is necessary to accurately predict long-term graft loss at the individual level (see below and Discussion).

**Table 3. Comparison of performance of four machine learning models for long-term graft loss prediction.**

| Method | Accuracy | Precision | Recall | F1 score | ROC-AUC |
|---|---|---|---|---|---|
| RF | 0.62 | 0.66 | 0.73 | 0.61 | 0.78 |
| Logistic Regression | 0.66 | 0.66 | 0.73 | 0.63 | 0.74 |
| XGBoost | 0.68 | 0.62 | 0.67 | 0.62 | 0.69 |
| Lightgbm | 0.68 | 0.63 | 0.68 | 0.63 | 0.69 |

## Stratification of living donor liver transplantation patients

In the prediction of early graft loss, the timing of graft loss in the patients was limited from 0 to 180 days after LDLT, whereas in the long-term graft loss prediction, the timing ranged from 186 to 7491 days. In other words, the low predictive performance was due not only to the period of data collection and the timing of prediction, but also to the high heterogeneity in the timing of graft loss. To address these limitations, we stratified the patients into smaller groups with reduced heterogeneity based on their clinical data and assessed the timing of graft loss for each stratified group. In the derivation cohort data, we attempted to stratify patients based on the result of the model for early graft loss prediction. We applied RF clustering (see Methods in detail) and identified five groups (i.e., G1: N = 31, G2: N = 81, G3: N = 161, G4: N = 106, G5: N = 184). **Fig 2A** indicates a two-dimensional Uniform Manifold Approximation and Projection (UMAP) embedding of the five stratified groups. Since we stratified the patients using a "supervised" RF classifier, we naturally identified a group (G1) of patients who lost the graft within 180 days after LDLT. Additionally, **Fig 2A** can be considered to represent the similarity (i.e., distance) between the early graft loss group (G1) and each of the other groups. Notably, the figure shows that G2 is located between G1 and the other groups (i.e., G3, G4, G5).

Next, we plotted and compared the graft survival curves for each group in **Fig 2B**. Interestingly, we found that the group most similar to G1 (i.e., G2) experienced graft loss significantly faster than most other groups (see **Fig 2B** and **Table D in S1 Text**). The median graft survival times were 101 days for G1 and 1106 days for G2 patients. While graft survival in G2 was significantly shorter than in G3, G4, or G5, there was no difference in survival time among G3, G4, and G5 patients. These results suggest that patients in G2, who have clinical characteristics similar to those in G1, experience graft loss relatively earlier than patients in G3, G4, and G5.

## Long-term prediction of group-level graft loss in living donor liver transplantation

We then attempted to evaluate the possibility of predicting long-term graft loss at the stratified group level. We first compared the stratified groups and found no statistically significant difference in survival time between G3, G4, and G5 (G3-G4: $p = 0.814$, G3-G5: $p = 0.3356$, G4-G5: $p = 0.369$, see **Table D in S1 Text** in detail). Therefore, hereafter, we merged G3, G4, and G5 into a combined group and focused on the following three groups for further analysis: early graft loss (i.e., **early-loss group**; corresponding to G1), intermediate graft loss (i.e., **intermediate-loss group**; corresponding to G2), and late or no graft loss (i.e., **late/no-loss group**; corresponding to the merged group of G3, G4, and G5).

To evaluate the possibility of long-term prediction of graft loss at the group level, we approached the problem as binary classification tasks, where the model aimed to predict whether a given patient belongs to early-loss or not (similarly for intermediate-loss and late/no-loss, see **Fig 2C**). For this purpose, we also evaluated RF, XGBoost, LightGBM, and logistic regression using cross-validation on the derivation cohort (see Methods for details). Among them, RF achieved the best overall performance, with ROC-AUCs of 0.96, 0.94, and 0.97 for the early-loss, intermediate-loss, and late/no-loss groups, respectively, and notably high recalls for the clinically critical early-loss (0.97) and intermediate-loss (0.98) groups (see **Fig 2D** and **Table 4**).These results implied a possibility of the long-term prediction of group-level graft loss in LDLT patients.

We also depicted 16 factors, arranged as the top 10 factors in each prediction without overlap, in **Fig 2E**, and compared each value among the groups in **Fig D in S1 Text**. As expected from the early graft loss prediction, the postoperative data are the most crucial factors for prediction among the SHAP values. Notably, we found that individuals classified in the intermediate-loss group were characterized by a greater amount of postoperative ascites on days 14 and 30 and lower PT%, similar to the early-loss group. However, in contrast to the early-loss group, the postoperative T-BIL in both donors and recipients in the intermediate-loss group were not as elevated, and intraoperative blood loss was comparatively lower. We describe the clinical characterization of G3, G4, and G5 in the Discussion.

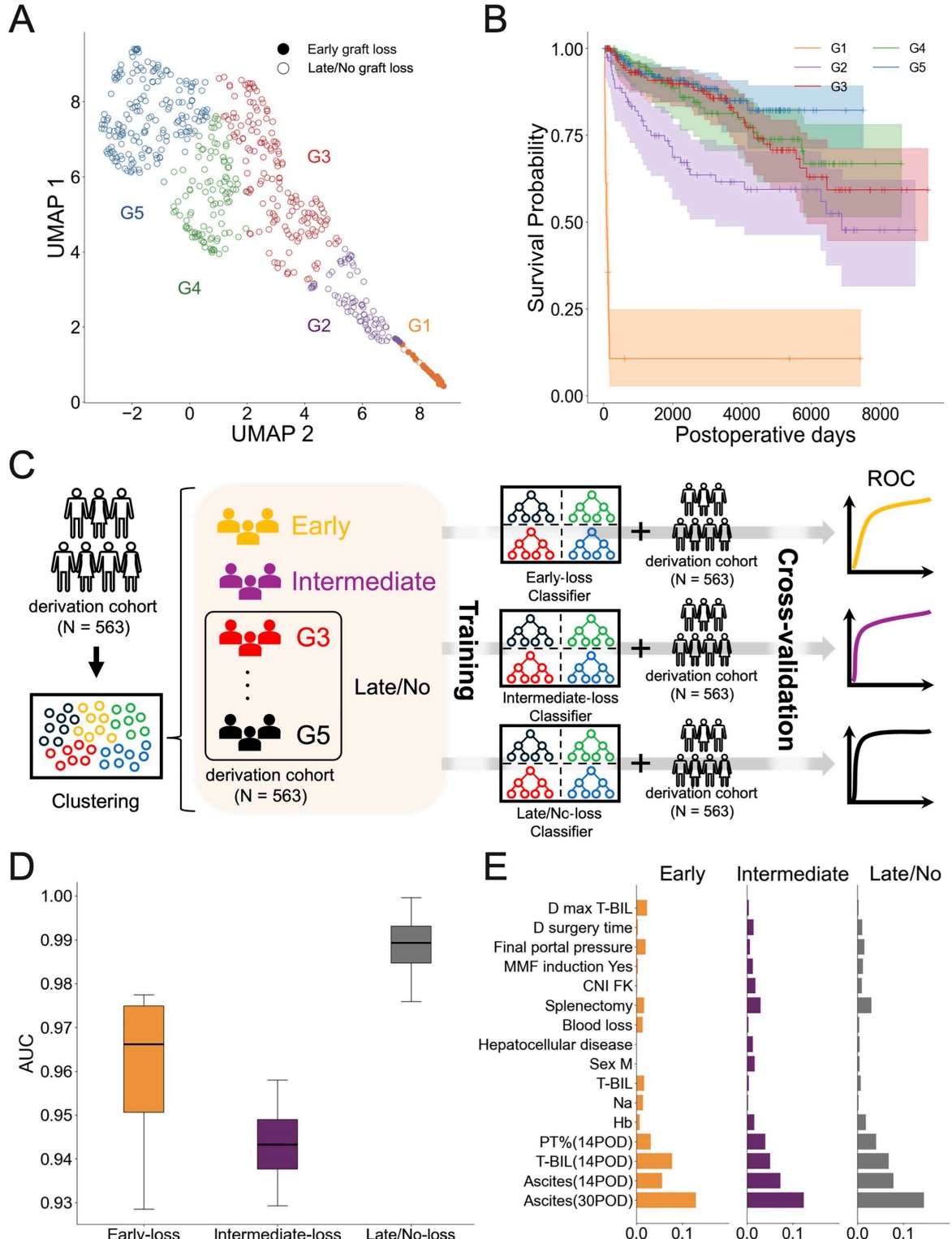

**Fig 2. Stratification and characterization of groups showing different graft survival: (A) UMAP visualization of the stratified derivation cohort data based on the distance matrix from the RF clustering.** Patients experiencing early graft loss are indicated by closed dots, and those without by open dots. **(B)** Survival curves for grafts for the stratified groups from **A** are plotted and color coded accordingly. Each shaded area around the curve

represents the 95% confidence interval, and short vertical lines on the curves indicate censored patients. **(C)** A workflow for developing and evaluating prediction models at the population level is shown. The derivation cohort data were stratified by RF clustering. For each group, RF binary classifiers were constructed, trained to predict whether a given patient belonged to the corresponding group, and were subsequently evaluated using a 5-fold cross-validation method. This figure was created by hand. **(D)** The AUCs calculated by ROCs of RF classifiers trained to predict whether each patient belongs to each group as a binary classification (e.g., either G1 or not) are plotted and colored accordingly. In each binary prediction, we obtained five AUC values and calculated the number of quartiles since we used a 5-fold cross-validation. **(E)** Feature importance of the predictive model in **D** is illustrated as a bar plot. The *x*-axis in each variable represents the mean SHAP value. Abbreviations and definitions: 30POD, 30 postoperative days; 14POD, 14 postoperative days; ascites, ascites volume; T-BIL, serum total bilirubin; PT%, prothrombin time activity percentage; Na, sodium; Hb, hemoglobin; blood loss, blood loss during surgery; splenectomy, the spleen was removed during surgery; hepatocellular disease, diagnosis on admission to hospital; CNI FK, calcineurin inhibitor was FK506 (tacrolimus); MMF induction yes, mycophenolate mofetil at the time of induction; final portal pressure, portal pressure measured at the end of surgery; D surgery time, the time of donor surgery; D max T-BIL, the maximum serum total bilirubin level of the donor within 30 days postoperatively.

**Table 4. Comparison of binary classifier performance among four machine learning models for early-loss, intermediate-loss, and late/no-loss groups.**

| Random forest | Early-loss vs other | Intermediate-loss vs other | Late/No-loss vs other | Macro Avg |
|---|---|---|---|---|
| ROC-AUC | 0.96 (0.02) * | 0.94 (0.01) | 0.99 (0.01) | 0.96 (0.02) |
| Precision | 0.46 (0.26) | 0.59 (0.09) | 0.99 (0.01) | 0.68 (0.28) |
| Recall | 0.97 (0.05) | 0.98 (0.02) | 0.97(0.01) | 0.97 (0.04) |
| F1 | 0.57 (0.21) | 0.73 (0.06) | 0.98 (0.01) | 0.76 (0.21) |
| Accuracy | 0.89 (0.08) | 0.89 (0.03) | 0.97 (0.02) | 0.92 (0.06) |
| **Logistic Regression** | **Early-loss vs other** | **Intermediate-loss vs other** | **Late/No-loss vs other** | **Macro Avg** |
| ROC-AUC | 0.93 (0.02) | 0.87 (0.01) | 0.97 (0.01) | 0.92 (0.04) |
| Precision | 0.37 (0.19) | 0.47 (0.08) | 0.98 (0.01) | 0.61 (0.29) |
| Recall | 0.93 (0.07) | 0.85 (0.05) | 0.91(0.07) | 0.90 (0.07) |
| F1 | 0.50 (0.16) | 0.60 (0.06) | 0.94 (0.03) | 0.94 (0.03) |
| Accuracy | 0.88 (0.06) | 0.83 (0.05) | 0.91 (0.05) | 0.68 (0.21) |
| **XGBoost** | **Early-loss vs other** | **Intermediate-loss vs other** | **Late/No-loss vs other** | **Macro Avg** |
| ROC-AUC | 0.95 (0.02) | 0.95 (0.02) | 0.99 (0.01) | 0.96 (0.02) |
| Precision | 0.53 (0.28) | 0.58 (0.03) | 0.99 (0.01) | 0.70 (0.26) |
| Recall | 0.90 (0.06) | 0.96 (0.04) | 0.94 (0.02) | 0.93 (0.05) |
| F1 | 0.63 (0.19) | 0.73 (0.02) | 0.97 (0.01) | 0.77 (0.18) |
| Accuracy | 0.93 (0.05) | 0.90 (0.01) | 0.95 (0.02) | 0.92 (0.03) |
| **Lightgbm** | **Early-loss vs other** | **Intermediate-loss vs other** | **Late/No-loss vs other** | **Macro Avg** |
| ROC-AUC | 0.96 (0.02) | 0.94 (0.02) | 0.99 (0.01) | 0.96 (0.02) |
| Precision | 0.46 (0.32) | 0.63 (0.07) | 0.995 (0.01) | 0.69 (0.29) |
| Recall | 0.97 (0.05) | 0.95 (0.04) | 0.94 (0.04) | 0.95 (0.05) |
| F1 | 0.55 (0.23) | 0.76 (0.06) | 0.97 (0.03) | 0.76 (0.22) |
| Accuracy | 0.88 (0.08) | 0.91 (0.03) | 0.95 (0.04) | 0.91 (0.06) |

*mean (standard deviation).

## Multi-class classification for long-term prediction of group-level graft loss in living donor liver transplantation

While binary classification of each group yielded high accuracy, such pairwise discrimination is more appropriate for subgroup characterization than for direct clinical application. Therefore, we next developed multi-class classifiers—RF, XGBoost, LightGBM, and logistic regression—to predict the specific group of each patient (i.e., early-loss, intermediate-loss, and late/no-loss). Among these, XGBoost achieved the best overall performance across groups (F1 scores: 0.57 for early-loss, 0.73 for intermediate-loss, and 0.97 for late/no-loss; see **Table 5**).

**Table 5. Comparison of multi-class classifier performance among four machine learning models for early-loss, intermediate-loss, and late/no-loss groups.**

| Random forest | Early-loss | Intermediate-loss | Late/No-loss | Macro avg |
|---|---|---|---|---|
| Precision | 0.69 (0.21)* | 0.69 (0.00) | 0.93 (0.02) | 0.77 (0.07) |
| Recall | 0.22 (0.10) | 0.62 (0.08) | 0.99 (0.00) | 0.61 (0.04) |
| F1 | 0.32 (0.11) | 0.65 (0.04) | 0.96 (0.01) | 0.64 (0.04) |
| Accuracy | 0.95 (0.01) | 0.90 (0.01) | 0.94 (0.02) | 0.90 (0.01) |
| **Logistic Regression** | **Early-loss** | **Intermediate-loss** | **Late/No-loss** | **Macro avg** |
| Precision | 0.49 (0.14) | 0.62 (0.10) | 0.95 (0.01) | 0.69 (0.08) |
| Recall | 0.52 (0.18) | 0.60 (0.06) | 0.95 (0.02) | 0.69 (0.06) |
| F1 | 0.50 (0.15) | 0.61 (0.07) | 0.95 (0.01) | 0.69 (0.07) |
| Accuracy | 0.94 (0.02) | 0.89 (0.02) | 0.92 (0.01) | 0.88 (0.02) |
| **XGBoost** | **Early-loss** | **Intermediate-loss** | **Late/No-loss** | **Macro avg** |
| Precision | 0.69 (0.21) | 0.74 (0.05) | 0.97 (0.02) | 0.80 (0.08) |
| Recall | 0.51 (0.11) | 0.74 (0.14) | 0.98 (0.01) | 0.74 (0.05) |
| F1 | 0.57 (0.12) | 0.73 (0.08) | 0.97 (0.01) | 0.76 (0.06) |
| Accuracy | 0.96 (0.02) | 0.93 (0.02) | 0.95 (0.02) | 0.92 (0.02) |
| **Lightgbm** | **Early-loss** | **Intermediate-loss** | **Late/No-loss** | **Macro avg** |
| Precision | 0.56 (0.07) | 0.75 (0.03) | 0.97 (0.02) | 0.76 (0.03) |
| Recall | 0.54 (0.08) | 0.69 (0.11) | 0.98 (0.01) | 0.74 (0.03) |
| F1 | 0.55 (0.05) | 0.71 (0.05) | 0.97 (0.01) | 0.74 (0.03) |
| Accuracy | 0.95 (0.01) | 0.92 (0.01) | 0.95 (0.02) | 0.91 (0.02) |

*mean (standard deviation).

Applying the multi-class XGBoost classifier trained on the derivation cohort to the validation cohort, we obtained predicted groups (early-loss*: N = 6, intermediate-loss*: N = 36, late/no-loss*: N = 143), whose survival curves are shown in Fig 3B. The three predicted groups exhibited distinct graft survival curves. However, none of the nine patients who experienced early graft loss in the validation cohort were classified as early-loss*. Consequently, statistical analysis did not detect a significant difference in survival time between early-loss* and intermediate-loss* (N.S.), whereas intermediate-loss* and late/no-loss* remained significantly different ($p < 0.05$; **Table G in** S1 Text).

Taken together with the cross-validation results in the derivation cohort, where recall and precision for early-loss (0.51 and 0.69, respectively) and intermediate-loss (0.74 and 0.74) were considerably lower than those for late/no-loss (0.98 and 0.97), these findings suggest that early-loss and intermediate-loss patients share highly similar characteristics, making them difficult to distinguish in multi-class classification (see Discussion).

### Hierarchical classification for long-term prediction of group-level graft loss in living donor liver transplantation

We found that the multi-class classification may not successfully distinguish between early-loss and intermediate-loss because of these groups' high similarity. This should be because each probability of the classifier's output is evaluated with equal weighting. Therefore, we then applied a hierarchical classification approach focused on accurately predicting early graft loss: the most important (highest risk of graft loss) population. Specifically, a binary early-loss classifier was first applied to predict whether a patient was early-loss* or not. We then identified late/no-loss* by applying the binary late/no-loss classifier (because of better performance than the intermediate-loss classifier, see **Fig 2D**) to the non-early-loss*

PLOS Computational Biology

population (i.e., not predicted as early-loss*). We note that intermediate-loss* refers to patients who were not classified into either early-loss* or late/no-loss* (Fig 3A, see Methods in detail).

As with the multi-class approach, we evaluated RF, XGBoost, LightGBM, and logistic regression. XGBoost again achieved the best performance in cross-validation of the derivation cohort, with improved performance compared to the

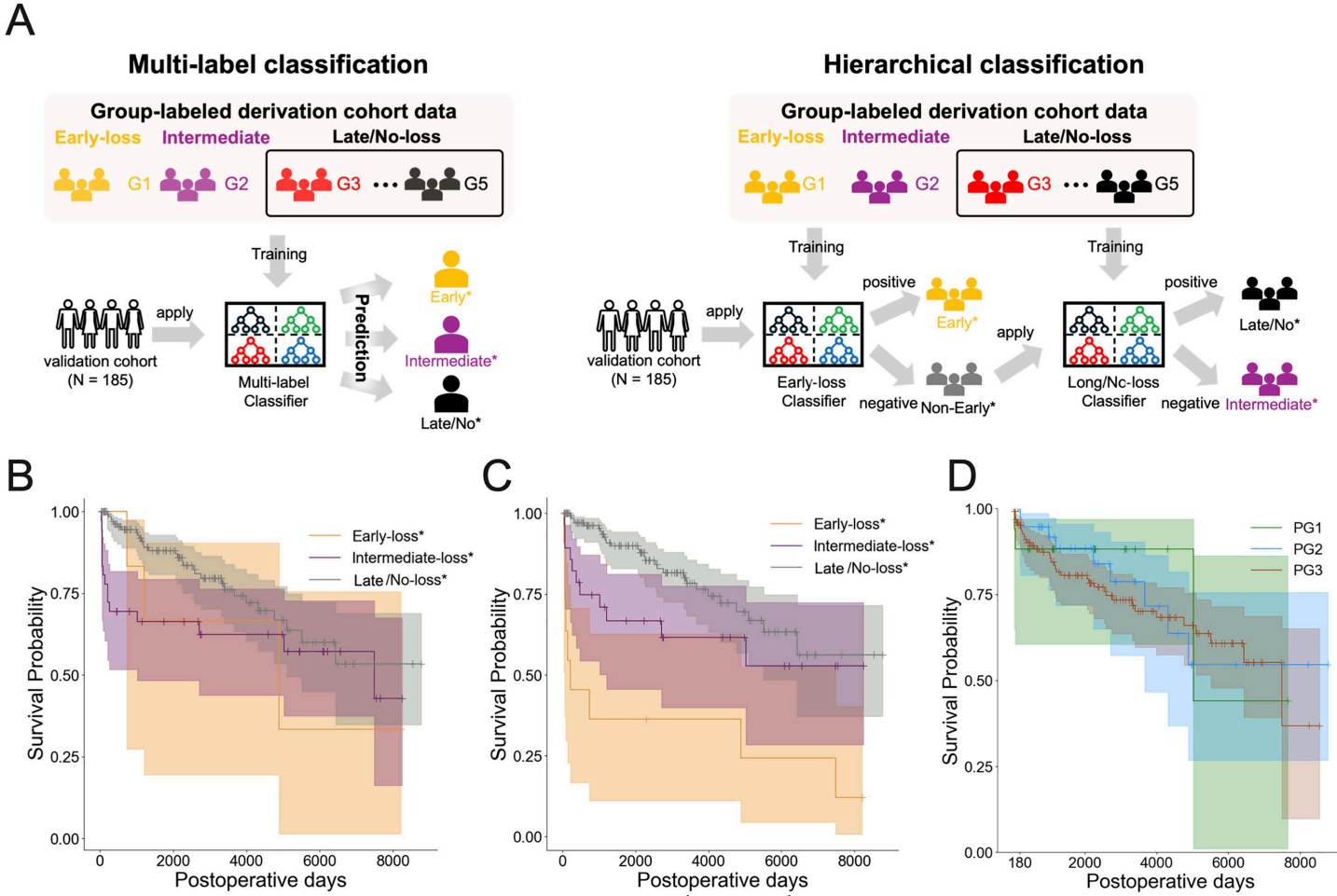

**Fig 3. Predicting stratified groups in the validation cohort: (A)** Workflows for the multi-class classification and hierarchical classification models are illustrated. For the multi-class classification, the multi-class classifier was constructed and trained to predict whether a given patient belonged to the corresponding group (i.e., early-loss, intermediate-loss, and late/no-loss) by using group-labeled derivation cohort data. The validation data were classified into three groups: early-loss*, intermediate-loss*, and late/no-loss*. In contrast, for hierarchical classification, early-loss and late/no-loss binary classifiers were constructed and trained to predict whether a given patient belonged to the corresponding group (i.e., early-loss and late/no-loss) by using group-labeled derivation cohort data. The validation data were then first classified into early-loss* or non-early-loss* using the early-loss classifier. Second, non-early-loss* patients were classified into late/no-loss* and others using the late/no-loss classifier. The remaining patients (i.e., others) were assigned to intermediate-loss*. This figure was created by hand. **(B)** Graft survival curves by predicted group in the validation cohort data by multi-class XGBoost classifier are plotted and colored accordingly. Each colored shaded area and short vertical line on the curve indicate the 95% confidence interval and a censored patient, respectively. **(C)** Graft survival curves by predicted group in the validation cohort data by the hierarchical XGBoost classifier are plotted and colored accordingly. Each colored shaded area and short vertical line on the curve indicate the 95% confidence interval and a censored patient, respectively. **(D)** Graft survival curves by classified group in the validation cohort data by predictive score are plotted and colored accordingly. PG1: Group with predictive score less than 1.15; PG2: Group with predictive score between 1.15 and 1.30; PG3: Group with predictive score of 1.30 or higher.

multi-class classification (early-loss: recall 0.71, precision 0.68; **Table 6**). We therefore applied the hierarchical XGBoost classifier to the validation cohort. Fig 3C shows the survival curves for each group of patients predicted by the hierarchical classification. These predicted groups (i.e., early-loss*: N = 6, intermediate-loss*: N = 47, late/no-loss*: N = 132) effectively distinguished the patients, resulting in statistically significant differences in survival time between all groups (i.e., early-loss* vs. intermediate-loss*: $p < 0.05$, early-loss* vs. late/no-loss*: $p < 0.001$, intermediate-loss* vs. late/no-loss*: $p < 0.05$; see **Table G in** S1 Text). Indeed, 56% of patients with early graft loss in the validation cohort data (5 of 9 patients) were classified as early-loss*.

Although and the early- and intermediate-loss groups are similar, they actually represent different groups in terms of survival time, and they can be somewhat distinguished on the basis of data up to 30 days post-surgery, e.g., more ascites after LDLT and lower serum total bilirubin on postoperative day 14 among patients in the intermediate-loss group (**Fig E** and **Table B in** S1 Text). These findings suggest that the early-loss group includes many patients who were critically sick preoperatively and who received a small-for-size graft. On the other hand, the intermediate-loss group contains many patients with cholestatic disease, many of whom also had splenomegaly, which may explain the greater ascites after LDLT. The different survival time between the early- and intermediate-loss groups may be explained as that 35.5% of the cause of death in the intermediate-loss group was recurrence of primary disease or de novo malignancy compared with 7.4% in the early-loss group.

Finally, we compared the hierarchical classification with our previously developed predictive scores [9]. We calculated the predictive scores for the validation cohort data and classified three groups: PG1, PG2, and PG3 (see Methods). Fig 3D shows the survival curves for PG1, PG2, and PG3, respectively. Our statistical analysis did not find any significant

**Table 6. Comparison of hierarchical classifier performance among four machine learning models for the early-loss, intermediate-loss, and late/no-loss groups.**

| Random forest | Early-loss | Intermediate-loss | Late/No-loss | Macro avg |
|---|---|---|---|---|
| Precision | 0.72 (0.01)* | 0.87 (0.02) | 0.96 (0.01) | 0.85 (0.02) |
| Recall | 0.68 (0.14) | 0.76 (0.12) | 0.99 (0.01) | 0.81 (0.17) |
| F1 | 0.69 (0.08) | 0.80 (0.06) | 0.97 (0.01) | 0.82 (0.13) |
| Accuracy | 0.97 (0.01) | 0.95 (0.02) | 0.95 (0.02) | 0.96 (0.02) |
| **Logistic Regression** | **Early-loss** | **Intermediate-loss** | **Late/No-loss** | **Macro avg** |
| Precision | 0.68 (0.01) | 0.82 (0.03) | 0.96 (0.02) | 0.82 (0.02) |
| Recall | 0.62 (0.21) | 0.77 (0.12) | 0.99 (0.01) | 0.79 (0.21) |
| F1 | 0.62 (0.09) | 0.79 (0.10) | 0.97 (0.01) | 0.80 (0.16) |
| Accuracy | 0.96 (0.01) | 0.94 (0.03) | 0.96 (0.02) | 0.95 (0.02) |
| **XGBoost** | **Early-loss** | **Intermediate-loss** | **Late/No-loss** | **Macro avg** |
| Precision | 0.68 (0.02) | 0.79 (0.02) | 0.97 (0.01) | 0.81 (0.02) |
| Recall | 0.71 (0.06) | 0.80 (0.09) | 0.98 (0.01) | 0.83 (0.13) |
| F1 | 0.69 (0.11) | 0.79 (0.07) | 0.97 (0.01) | 0.82 (0.14) |
| Accuracy | 0.96 (0.02) | 0.94 (0.02) | 0.95 (0.01) | 0.95 (0.02) |
| **LightGBM** | **Early-loss** | **Intermediate-loss** | **Late/No-loss** | **Macro avg** |
| Precision | 0.67 (0.02) | 0.84 (0.02) | 0.96 (0.01) | 0.83 (0.02) |
| Recall | 0.71 (0.12) | 0.80 (0.10) | 0.99 (0.01) | 0.83 (0.15) |
| F1 | 0.66 (0.08) | 0.81 (0.06) | 0.98 (0.01) | 0.82 (0.14) |
| Accuracy | 0.96 (0.02) | 0.95 (0.02) | 0.96 (0.01) | 0.96 (0.02) |

*mean (standard deviation).

differences among the groups. Overall, our hierarchical classification demonstrates better performance in predicting long-term graft loss at the group level in LDLT patients.

## Discussion

Previous observational studies, such as those using predictive scores [8,9], D-MELD [7], TB-INR criteria [10], and the eGLR score [11], have aimed to predict early graft loss but have faced limitations. First, the simplicity of these criteria and models [7,10] restricts their predictive performance. Second, their general applicability is unclear, as the performance of the validation data was not verified [8,9]. In this study, using extensive clinical data collected from 748 patients who underwent LDLT in Kyushu University Hospital between 1997 and 2023, we analyzed clinical variables associated with whether the patients lost the transplanted graft. To ensure generalizability, the data were divided into derivation and validation cohorts, and machine learning was applied to capture subtle and complex predictors of graft loss. Specifically, we evaluated the prediction of graft loss at two levels: the individual patient level and the stratified group level.

For individual-level predictions, we achieved an exceptionally high accuracy rate of prediction of early graft loss—defined as loss within 180 days after LDLT—using only 6 variables collected up to 30 days postoperatively (ROC-AUC = 0.90). In contrast, the prediction of long-term graft loss, defined as loss beyond 180 days, was less accurate due to the wide variability in timing (186–7491 days) and high heterogeneity of patient characteristics. While donor and graft characteristics (e.g., donor age, graft weight) have been highlighted in prior work [12,13], our model demonstrated strong performance using only recipient-related perioperative and postoperative variables. These findings suggest that in a high-volume center with standardized donor selection and operative protocols, donor- or graft-related variables may offer limited discriminative value, whereas recipient condition and early postoperative trajectory are more critical.

For group-level predictions, unlike previous observational studies on LDLT that relied on simple univariate or multivariate analyses to predict early graft loss [6,7,9–11], our study focused on stratifying graft survival outcomes. We successfully stratified the highly heterogeneous LDLT patient population into five distinct clinical groups (G1-G5) based on graft prognosis. Remarkably, 90% of early graft losses were concentrated in G1 (early-loss), whereas all patients in G3, G4, and G5 (late/no-loss) demonstrated graft survival beyond 180 days. Although only 10% of early graft losses were included in G2 (intermediate-loss), graft survival in this group was significantly shorter than in the late/no-loss group. By reducing patient heterogeneity through stratification, binary classification achieved high ROC-AUCs of 0.96, 0.94, and 0.99 for the early-loss, intermediate-loss, and late/no-loss subgroups, respectively. Although multi-class classification was more challenging, a hierarchical approach enabled classification of validation cohort patients into three clinically distinct groups with different survival profiles.

A key finding was the identification of the intermediate-loss subgroup, which shared perioperative features with the early-loss group but exhibited graft failure at a later stage. The resemblance of these groups, together with their distinction from the late/no-loss group, shaped the performance of hierarchical prediction for early graft loss. In addition, the mechanisms of graft failure appeared distinct. Early graft loss was associated with higher early postoperative T-BIL and intraoperative blood loss, pointing to perioperative injury or small-for-size physiology. By contrast, intermediate graft loss reflected initially stable recovery followed by complications such as biliary stricture, recurrent primary disease (e.g., primary sclerosing cholangitis or HCC), or de novo malignancy. Indeed, 14 of 81 patients (17.3%) developed biliary strictures and 6 ultimately lost the graft after treatment failure. These findings highlight the importance of identifying intermediate-loss as a distinct group in which targeted monitoring and timely intervention may improve outcomes.

The remaining groups, G3, G4, and G5, were characterized by different clinical profiles. As shown by SHAP analysis, G3 was characterized by greater blood loss and lower serum sodium levels, G4 by lower PT% (postoperative 14 days) and an intermediate T-BIL among the late/no-loss subgroup, and G5 by the proportion of splenectomies and mycophenolate mofetil at the time of induction (see **Fig E and Tables B** and **C in** S1 Text). Comparisons confirmed that G3 patients had worse postoperative courses, while G5 patients had the most favorable conditions.

Our study does have several limitations. First, our analysis was based on data from 748 LDLT patients at a single high-volume center. Although this cohort is relatively large compared with prior LDLT studies [8–11], it remains modest, particularly after division into derivation and validation datasets. The validation cohort contained only 185 patients, including 9 early graft loss patients (4.9%), resulting in a marked class imbalance that may have influenced predictive performance. To mitigate this, we applied the synthetic minority over-sampling technique (SMOTE) to the derivation data, while all evaluations were performed on real, unaugmented validation data to avoid performance inflation. We further confirmed robustness through internal cross-validation and sensitivity analyses using different random seeds (see Methods and **Fig A in** S1 Text), which consistently demonstrated stable performance and supported internal validity. Nevertheless, external validation using data from independent transplant centers is essential to confirm the generalizability of our findings. Multi-center datasets, however, bring their own challenges. For instance, variables such as ascites may be prone to subjective bias when collected across institutions. At the same time, existing machine learning models developed using multi-center datasets have often focused on DDLT populations [14,15], limiting their direct applicability to LDLT. In this context, our study not only pioneered the development of machine learning models in the growing field of LDLT research [2], using homogeneous, single-center data, but also provided a more nuanced understanding of the factors contributing to early graft loss, including physician-interpreted features like ascites. Future work should balance these trade-offs by integrating larger and more diverse LDLT cohorts.

Second, our analysis only used data collected within 30 days after LDLT, which limits the accuracy of predicting graft survival beyond 180 days. This constraint, combined with the wide heterogeneity in long-term outcomes, reduced model performance for late graft loss. Incorporating longitudinal data collected beyond 30 days could improve prediction, while multimodal information such as donor-recipient familial relationships or omics-based biological connections using graph-based learning approaches [16] may further refine models. Furthermore, important confounders—including socioeconomic status, nutritional condition, and long-term immunosuppressive management—were unavailable in our dataset. Future studies should integrate such multimodal data to strengthen the robustness and clinical applicability of predictive models.

Finally, institutional and temporal factors may have influenced our analysis. At Kyushu University, extended left lobe grafts and simultaneous splenectomy with portocaval shunt ligation for portal flow modulation are more frequently performed than at other high-volume LDLT centers [17], which may limit generalizability. In addition, the influence of the learning curve is unavoidable because the cases analyzed occurred over a long period [18]. Surgical practices also evolved over time, including an increased use of right lobe grafts from April 2004 and a change in hepatic vein reconstruction in April 2011 [19]. Nonetheless, no significant temporal clustering was observed across the groups (i.e., early-loss, intermediate-loss, and late/no-loss), minimizing concern for model bias. Immunosuppressive protocols remained consistent throughout the study, which may explain the absence of drug-related predictors of early graft loss. External validation in centers with different surgical practices will be important to confirm the robustness of these findings.

In conclusion, this study is the first to stratify LDLT recipients into five groups with different graft survival profiles and to identify an intermediate-loss subgroup characterized by earlier graft loss and unique clinical phenotypes. Notably, these stratified groups could be predicted using clinical data up to 30 days postoperatively, allowing for timely prognostication and potential planning for re-transplantation before the onset of severe infection.

## Methods

### Ethics statement

We retrospectively reviewed data for 748 patients who underwent primary adult LDLT at Kyushu University Hospital between 1997 and 2023. The study protocol was approved by the institutional review board of Kyushu University Hospital (2022–146) and was conducted according to the Declaration of Helsinki. Given the retrospective nature of this study and the use of anonymized data to prevent identification of individual patients, written informed consent was not obtained.

Additionally, some patients had passed away during the study period. Instead, informed consent was obtained through an opt-out process in accordance with institutional guidelines.

## Study data

The criteria for performing LDLT on patients without HCC were: (1) absence of other potentially curative treatments; and (2) absence of other organ dysfunction. No age restriction was applied to recipients. Criteria for patients with HCC were: (1) absence of other potentially curative treatment; (2) no extrahepatic metastasis; and (3) no major vascular infiltration. Living donors were selected from volunteers [20] and were required to be within three degrees of consanguinity or the spouse of the recipient, aged between 20 and 65 years. For donors not within three degrees of consanguinity with the recipient, individual approval was obtained from the Ethics Committee of Kyushu University Hospital. Good Samaritan donation was not used. Three-dimensional computed tomography was performed for volumetric analysis and delineation of vascular anatomy. Decisions about graft types were based on the preoperatively predicted graft weight to standard liver weight (GW/SLW) ratio. Left lobe + caudate lobe grafts were basically used when the preoperatively predicted GW/SLW ratio was ≥ 35%, but a relatively smaller graft, such as a GW/SLW ratio between 30% and 35%, was selected when the donor was younger than 30 years of age [20]. The indications for splenectomy (Spx) at our center were as follows:

1. Preoperative low predicted graft weight (GW/SLW) ratio (35% or less);

2. Portal hypertension indicated by a large portosystemic shunt, splenomegaly, or risky esophago-gastric varices;

3. High portal pressure (above 20 mmHg) after unclamping [17,21].

Note that bile duct reconstruction has been performed by duct-to-duct anastomosis as much as possible since 2001 [22].

Perioperative management of recipients, including immunosuppression regimens, has been described previously [17]. Immunosuppression was initiated using a protocol based on either tacrolimus (Prograf; Astellas Pharma, Tokyo, Japan) or cyclosporine A (Neoral; Novartis Pharma K.K., Tokyo, Japan) with steroids. The target trough concentration for tacrolimus was set at 10 ng/mL for 3 months after LDLT, followed by 5–10 ng/mL. The target trough concentration for cyclosporine A was set at 250 ng/mL for 3 months after LDLT, followed by 150–200 ng/mL. Mycophenolate mofetil was used, beginning with 2 or 3 g/day on the day after LDLT; the dose was tapered and discontinued 6 months after LDLT. Trough concentrations of mycophenolate mofetil were not measured.

## Data preparation

We excluded data from 19 patients who did not have data up to 30 days postoperatively, incorporating 46 numerical variables and 37 categorical variables as detailed in **Table A in** S1 Text. Each variable had less than 30% missing values, as we excluded data with more than 30% missing values. Additionally, we converted categorical data into dummy variables. We selected 563 of the total 748 patients (75%) for the derivation cohort to develop the model and assigned 185 patients (25%) to the validation cohort. All data, encompassing the 46 numerical and 105 categorical dummy variables, underwent standardization and imputation using k-Nearest Neighbors (Python scikit-learn) to address missing values. All procedures were carried out solely within the derivation cohort, and the fitted scaler and imputer obtained from the derivation cohort data were then applied to the validation cohort data. In addition, we evaluated alternative imputation strategies such as multiple imputation by chained equations (MICE) and random forest–based imputation. These methods yielded comparable predictive performance, indicating that the choice of imputation method did not substantially affect model accuracy.

## Survival time analysis

The Kaplan-Meier method is used for graft survival analysis. We used KaplanMeierFitter from Python's lifelines. When necessary, survival times were compared among different groups using the Generalized Wilcoxon test with Benjamini-Hochberg correction (R survminer, version 4.2.0, **Tables D** and **G in** S1 Text).

## Comparison of machine learning methods

We developed five distinct classification models to predict graft outcomes: (1) early graft loss, (2) long-term graft loss, (3) binary classification for each of the three stratified groups (early-loss, intermediate-loss, and late/no-loss), (4) multi-class classification, and (5) hierarchical classification. For each classification task, four machine learning algorithms were applied: logistic regression (Python, scikit-learn), RF (Python, scikit-learn), XGBoost (Python, xgboost), and LightGBM (Python, lightgbm).

Predictive performance for binary classification tasks was assessed using ROC-AUC, precision, recall, and F1 score. For multi-class and hierarchical classification tasks, ROC-AUC was not applicable; therefore, performance was evaluated using precision, recall, and F1 score only. The model achieving the best overall performance for each task was selected accordingly.

## Hyperparameter tuning

In each classification task, hyperparameter tuning was performed exclusively within the derivation cohort using 4-fold cross-validation with GridSearchCV (scikit-learn, Python). All predefined combinations of hyperparameter values were exhaustively evaluated, and the configuration yielding the highest cross-validated performance was selected as optimal.

In RF, XGBoost, and LightGBM, the following parameter grids were used:

- Random Forest: $n_{estimators} \in [10, 50, 100, 500, 1000]$, $max_{depth} \in [None, 10, 20]$, $min\_samples\_split \in [2, 5, 10]$.

- XGBoost: $n_{estimators} \in [10, 50, 100, 500, 1000]$, $max_{depth} \in [None, 10, 20]$, $min_{child_{weight}} \in [1, 2]$, $eta \in [0.01, 0.1, 1.0]$, $gamma \in [0, 0.1]$.

- LightGBM: $n_{estimators} \in [10, 50, 100, 1000]$, $max_{depth} \in [4, 8, 16]$, $num_{leaves} \in [31, 15, 7, 3]$, $learning\_rate \in [0.1, 0.05, 0.01]$.

Logistic regression was implemented with default settings in scikit-learn, and no hyperparameter tuning was performed.

Importantly, the validation cohort was not used at any stage of hyperparameter tuning to avoid information leakage. Once the optimal hyperparameters were determined, the best-performing model for each classification task was re-trained on the full derivation cohort and evaluated on the validation cohort.

## Supervised machine learning for predicting early graft loss and stratifying patients

As described in the previous section, we aimed to identify the most effective model for predicting early graft loss. Predictive performance was comparable among the algorithms evaluated (**Table 1**); however, RF was selected for downstream stratification because of its ensemble structure based on independently constructed trees via bootstrap aggregation, which is better suited for clustering than the sequential nature of boosting models.

Early graft loss was framed as a binary classification task, and RF was implemented using the RandomForestClassifier from Python's scikit-learn. The model was trained on the derivation cohort using all available features—preoperative, intraoperative, and postoperative variables from both recipients and donors (**Table A in** S1 Text). Model performance was evaluated on the validation cohort using ROC-AUC. Precision, recall, and F1 score were calculated based on the threshold determined by the Youden index.

To address class imbalance, we applied SMOTE (Python, Imbalanced-learn) [23] to the training data during model development. No synthetic data were included in the validation phase to ensure an unbiased performance evaluation.

After an RF dissimilarity (i.e., the distance matrix between all pairs of samples) was obtained, it was visualized with UMAP on a two-dimensional plane and hierarchized with Spectral Clustering (Python scikit-learn). The optimal number of clusters was determined by the eigengap heuristic method.

## Comparison with other models and criteria for early graft loss

We compared the performance of our RF model to predict early graft loss with the predictive score, D- MELD score, TB-INR criteria, and eGLR score (see **Table 2**). These models were applied exclusively to the validation cohort data.

We calculated the predictive score in the validation cohort data by using the following equation [9]:

$$\text{Predictive score} = 0.011 \times (\text{GW}(\%)) - 0.016 \times (\text{Donor age (years)}) \times 0.008 \times (\text{MELD score}) - 0.15 \times (1 \text{ if Shunt exists, else } 0) + 1.757$$

For the D-MELD score, we simply calculated the product of the preoperative recipient's MELD score and the donor's age in the validation cohort data [7].

For TB-INR criteria [10], we applied the criteria to 113 patients who had values for both serum total bilirubin and pro-thrombin time international normalized ratio (PT-INR) on postoperative day 7 in the validation cohort data (N = 18).

We calculated the eGLR score in the validation cohort data by using following equation [11]:

$$(\text{eGLR score}) = 10 \times (0.054 \times (\text{Donor age (years)}) + 1.851 \times (1 \text{ if GW(g)} < 610 \text{ else } 0) + 0.128 \times (\text{MELD score})) - 8.128 - 8.128$$

## Feature selection for early graft loss prediction

As for early graft loss prediction, RF with optimized hyperparameters was used. To mitigate potential confounding in feature importance, a correlation matrix was computed prior to feature selection, and variables with pairwise correlations greater than 0.7 were excluded.

We divided the derivation cohort data into training and testing data for 4-fold cross-validation. Then, we adapted the training data to the RF model, calculated SHAP values (Python, shap) [24–27], and ordered the variables according to their contribution to the prediction. We trained an RF model using only one variable with the highest contribution and tested the prediction performance by ROC-AUC on the test data. The number of variables used to train the RF model was then increased in the order of highest contribution, and the variables with the highest ROC-AUC were recorded. This process was repeated five times by cross-validation, and variables recorded in two or more of the five folds were considered important variables (**Fig 1D**).

Feature selection was conducted exclusively within the derivation cohort. Finally, an RF model was trained on the full derivation cohort data using only the selected important variables and the optimized hyperparameters, and its performance was evaluated on the validation cohort (**Fig 1E**).

## Supervised machine learning for predicting long-term graft loss

For long-term graft loss prediction, we similarly evaluated multiple machine learning algorithms (i.e., logistic regression, RF, XGBoost, and LightGBM,) to identify the best-performing model. RF, XGBoost, and LightGBM demonstrated comparable predictive performance (**Table 3**). However, considering methodological consistency with the early graft loss RF prediction and to facilitate direct comparison, RF was selected for this prediction.

## Binary classifiers for characterizing stratified groups

We similarly evaluated multiple machine learning algorithms (i.e., logistic regression, RF, XGBoost, and LightGBM,) to identify the best-performing model. In this classification task, we approached the problem as three independent binary classification tasks, where the model aimed to predict whether a given patient belonged to early-loss or not (similarly for intermediate-loss and late/no-loss, respectively). Then we used RF because it achieved the best performance in macro-average ROC-AUC and Recall.

We used a 4-fold cross-validation method (StratifiedKfold, Python, scikit-learn) because only derivation cohort data have group labels (early-loss, intermediate-loss, and late/no-loss) in this tasks. Specifically, we split the derivation cohort data into four parts, used each for testing, and calculated metrics (**Table 4** and **Fig 2D**). Regarding the issue of data imbalance in the classifier, and evaluation of the significant variables, we applied the same approach (i.e., SMOTE and SHAP) described above for early graft loss prediction.

Importantly, in each fold of the cross-validation, preprocessing steps (scaler, imputer, and SMOTE) were applied only to the training data, not the testing data. In addition, no hyperparameter tuning was performed in this setting; all algorithms were used with default parameters.

### Multi-class classification

We approached long-term group-level prediction as a multi-class classification task, aiming to assign each patient to one of three groups (early-loss, intermediate-loss, and late/no-loss). Four machine learning algorithms (logistic regression, RF, XGBoost, and LightGBM) were compared to identify the best-performing model. Similarly, because only the derivation cohort had group labels, we applied stratified 4-fold cross-validation (StratifiedKFold, Python, scikit-learn) within this dataset. In each fold, scaling, imputation, and model training were performed exclusively on the training data and subsequently applied to the test data. No hyperparameter tuning was conducted during this stage. Among the four models, XGBoost achieved the highest macro-averaged performance across all metrics in the derivation cohort data (**Table 5**).

After evaluating the algorithms (XGBoost), we re-trained the multi-class XGBoost classifier on the entire derivation cohort and then applied it to the validation cohort. Each patient was then assigned to the group with the highest probability in the three outputs (i.e., the probabilities of early-loss*, intermediate-loss*, and late/no-loss*).

### Hierarchical classification

We first trained binary classifiers for early-loss and late/no-loss using derivation data, with only the six important factors identified in **Fig 1D** to focus on early-loss prediction. Patients classified as early-loss* by the first early-loss classifier were directly annotated as early-loss*, while the second late/no-loss classifier was applied to the remaining patients to identify late/no-loss*. Patients negative for both classifiers were defined as intermediate-loss*.

Four machine learning models (i.e., logistic regression, RF, XGBoost, and LightGBM) were compared to identify the best-performing model. Similarly, we applied stratified 4-fold cross-validation (StratifiedKFold, Python, scikit-learn) within this dataset. In each fold, scaling, imputation, and model training were performed exclusively on the training data and subsequently applied to the test data. No hyperparameter tuning was conducted during this stage. Among the four models, RF, XGBoost, and LightGBM demonstrated comparable predictive performance (**Table 6**). Then, considering methodological consistency with multi-class XGBoost prediction and to facilitate direct comparison, XGBoost was selected for this prediction. Again, the hierarchical XGBoost classifier (i.e., two binary classifiers for early-loss and late/no-loss) was re-trained on the entire derivation cohort and applied to the validation cohort.

### Statistical analysis

In **Figs 1F**, **1D**, **and 1E**, and **Tables C**, **E**, and **F** in S1 Text, variables were compared among different groups using pairwise Fisher's exact test with Bonferroni correction for categorical variables and analysis of variance with Bonferroni-corrected Wilcoxon rank-sum test for numerical variables. In **Table A in** S1 Text, variables were compared using Fisher's exact test for categorical variables and analysis of variance with Wilcoxson rank-sum test for numerical variables.
In **Table B in** S1 Text, analysis of variance with Kruskal-Wallis test for numerical variables was used. All statistical analyses were performed using R (version 4.2.0).

## Supporting information

**S1 Text.** **Fig A.** Sensitivity analysis of Random Forest. **Fig B.** Calibration plot and decision curve analysis for early graft loss prediction. **Fig C.** Early graft loss prediction without donor information. **Fig D.** Comparison of the important features between groups with early, intermediate, and late or no graft loss. **Fig E.** Comparison of important features between all groups. **Table A.** Comparison of clinical data between the derivation and validation datasets. **Table B.** Comparison of derivation data in each group. **Table C.** Comparison of clinical data in each group. **Table D.** Comparison of Generalized Wilcoxon test results with Benjamini-Hochberg correction in derivation cohort data. **Table E.** Comparison of Bonferroni-corrected Wilcoxon rank-sum test results. **Table F.** Comparison of Bonferroni-corrected Fisher's exact test results. **Table G.** Comparison of Generalized Wilcoxon test results with Benjamini-Hochberg correction in group-annotated validation cohort data.
(DOCX)

## Author contributions

**Conceptualization:** Naotoshi Nakamura, Shingo Iwami, Tomoharu Yoshizumi.

**Data curation:** Raiki Yoshimura, Takeru Matsuura, Takeo Toshima.

**Formal analysis:** Raiki Yoshimura, Takeru Matsuura.

**Investigation:** Naotoshi Nakamura.

**Methodology:** Raiki Yoshimura, Takeru Matsuura.

**Project administration:** Shingo Iwami, Tomoharu Yoshizumi.

**Resources:** Takeo Toshima, Tomoharu Yoshizumi.

**Supervision:** Naotoshi Nakamura, Takasuke Fukuhara, Kazuyuki Aihara, Katsuhito Fujiu, Shingo Iwami, Tomoharu Yoshizumi.

**Validation:** Naotoshi Nakamura, Kazuyuki Aihara, Katsuhito Fujiu.

**Visualization:** Takeru Matsuura.

**Writing – original draft:** Raiki Yoshimura, Takeo Toshima.

**Writing – review & editing:** Naotoshi Nakamura, Takeru Matsuura, Takasuke Fukuhara, Kazuyuki Aihara, Katsuhito Fujiu, Shingo Iwami, Tomoharu Yoshizumi.

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
