## [Decision Letter · Decision Letter 0]

30 May 2025

PCOMPBIOL-D-25-00261

Prediction of graft loss in living donor liver transplantation during the early postoperative period

PLOS Computational Biology

Dear Dr. Iwami,

Thank you for submitting your manuscript to PLOS Computational Biology. After careful consideration, we feel that it has merit but does not fully meet PLOS Computational Biology's publication criteria as it currently stands. Therefore, we invite you to submit a revised version of the manuscript that addresses the points raised during the review process.

Please submit your revised manuscript within 60 days Jul 30 2025 11:59PM. If you will need more time than this to complete your revisions, please reply to this message or contact the journal office at ploscompbiol@plos.org. Please include the following items when submitting your revised manuscript:

We look forward to receiving your revised manuscript.

Kind regards,

Hao Hu, Ph.D

Academic Editor

PLOS Computational Biology

Thomas Leitner

Section Editor

PLOS Computational Biology

**Journal Requirements:**

At this stage, the following Authors/Authors require contributions: Raiki Yoshimura, Naotoshi Nakamura, Takeru Matsuura, Takeo Toshima, Takasuke Fukuhara, Kazuyuki Aihara, Katsuhito Fujiu, Shingo Iwami, and Tomoharu Yoshizumi. Please ensure that the full contributions of each author are acknowledged in the "Add/Edit/Remove Authors" section of our submission form.

5) We notice that your supplementary Figures, and Tables are included in the manuscript file. Please remove them and upload them with the file type 'Supporting Information'. Please ensure that each Supporting Information file has a legend listed in the manuscript after the references list.

6) Please ensure that the funders and grant numbers match between the Financial Disclosure field and the Funding Information tab in your submission form. Note that the funders must be provided in the same order in both places as well. State what role the funders took in the study. If the funders had no role in your study, please state: "The funders had no role in study design, data collection and analysis, decision to publish, or preparation of the manuscript.".

**Reviewers' comments:**

Reviewer's Responses to Questions

**Comments to the Authors:**

Reviewer #1: The authors predict graft loss in living donor liver transplantation. This is an interesting interdisciplinary piece of work.

As I am ML specialist, my review will touch ML issues primarily. I consider the ms almost ready to publish, however, I would like to double-check, that the cross validation proceedure was done correctly. I assume it was, but want to be 100% sure and therefore ask for clarification.

Please confirm and clearly write in the ms, that only data from the training set was used for standardization, imputation, hyperparameter tuning etc. In the ms the training data set is denoted as derivation data set.

If data from the validation data set is used in any of standardization, imputation, hyperparameter tuning etc then the results would be flawed.

In line 209 - 211 The sentence: 'Note that in this model ... ' is unclear in this respect.

l 527 ' then the best parameters...'

typo l 432 computed tomography

Finally, I have a question. In order to use the model in practice, it is important that for an individual patient the prediction is correct. Could you perhaps discuss, whether the model allows to assess uncertainty so that a wrong classification can be avoided...

Reviewer #2: This study retrospectively analyzed data from 748 LDLT patients and applied machine learning to predict early graft loss within 180 days postoperatively, outperforming traditional models. A hierarchical prediction approach was developed to identify high- and intermediate-risk patients (G1 and G2) within 30 days after surgery, enabling timely clinical interventions to improve transplant outcomes.

The authors have done a lot of work in the field of living donor liver transplantation, but my main points of view are as follows.

1. In the “Significance Statement,” it is recommended to further emphasize the practical implications of this study for clinical decision-making, particularly how the model can guide early interventions.

2. In the “Main Text,” it would strengthen the rationale for developing LDLT-specific predictive models to include a brief literature review comparing the mechanisms and postoperative risks of LDLT versus DDLT.

3. The data presented in Table S4 do not conclusively demonstrate that the RF classifier is the optimal model, as its performance is not consistently superior to the others. We recommend including more detailed descriptions of the cross-validation procedures to enhance the rigor of the model selection process.

4. Although seven key predictors were identified by SHAP analysis, we suggest specifying clinical threshold values or abnormal ranges for each variable to improve interpretability and applicability in clinical settings.

5. The statement “however, we found that donor and graft information did not significantly contribute to the prediction of early graft loss” warrants further discussion. We recommend exploring potential clinical explanations for this finding.

6. The analysis indicating that G2 closely resembles G1 in characteristics but exhibits different survival outcomes would benefit from mechanistic insights into the clinical distinctions between these two groups.

7. In the “Group-Level Prediction” section, we recommend including confusion matrices or more comprehensive classification metrics such as F1-scores to better assess model performance across the three binary classification tasks.

8. We suggest a more in-depth discussion of the reasons behind the poor performance of the multilabel classification model, including specific evidence that low recall for G1 may be due to the small sample size.

9. To enhance the argument for this study’s contribution, we recommend a systematic comparison of the proposed model’s performance with existing LDLT prediction models, ideally supported by statistical significance testing.

10. As the discussion mentions that changes in surgical techniques and immunosuppressive regimens may affect prediction accuracy, we suggest including corresponding trend plots or descriptive analyses of these variables over time.

11. Regarding future work, please discuss the possibility of applying advanced graph learning techniques (10.1109/TNSE.2024.3524077) for improved performance.

Reviewer #3: The study is based on data from a single, high-volume center (Kyushu University Hospital), which may limit its generalizability. Variations in patient demographics, surgical practices, and perioperative management across institutions could affect the model’s performance elsewhere. Additionally, the external validation cohort is relatively small, and the number of early graft loss events (n=12 in the validation set) is particularly limited. This may compromise the statistical power and robustness of performance evaluation for the classification models, especially for rare outcomes. However, other concerns are included below.

1- Further clarification is needed on how the identified features can be interpreted clinically. For instance, guidance on actionable thresholds or how clinicians should adjust care based on model outputs is minimal.

2- The prediction of long-term graft loss using only early postoperative data showed limited success (ROC-AUC 0.59–0.78). The authors acknowledge this but might further elaborate on strategies to improve this (e.g., incorporating longitudinal follow-up data). Add calibration plots and decision curve analysis to evaluate clinical utility beyond ROC-AUC.

3- The grouping into G1–G5 and then merging into G1, G2, and G3+G4+G5 may be confusing to readers. A more intuitive naming convention or visual summary could improve clarity.

4- The authors uses k-NN imputation for missing values. Given the high dimensionality, the choice and validation of this method need more justification, as different imputation strategies might influence model performance.

5- Improve the discussion of model limitations and future directions, particularly for real-world deployment.

6- The absence of true external validation (e.g., from an independent institution or dataset) restricts the model’s demonstrated generalizability.

7- The number of early graft loss events (especially in the validation set) is quite small relative to the number of features considered, even after SHAP-based reduction. This raises concerns about overfitting and the reliability of model performance metrics.

8- Important confounders, such as socioeconomic factors, nutritional status, or long-term immunosuppressive management, were not mentioned or adjusted for. These may impact graft loss outcomes and should be considered or at least discussed.

9- While the manuscript mentions use of GridSearchCV for hyperparameter optimization, it lacks detail on the grid ranges tested, scoring metrics used for selection, or regularization strategies (e.g., cross-validation techniques to avoid overfitting).

10- SMOTE was used, but the manuscript lacks a discussion on potential pitfalls of synthetic data generation, such as introducing noise or inflating minority class performance unrealistically.

11- Further clinical interpretation—especially why some features (like preoperative home status or drainage ascites) are strong predictors—is underdeveloped.

12- Several key tables and figures critical to understanding the results are placed in the supplementary section (e.g., Tables S1–S6, Fig S1–S3). Consider moving the most important ones to the main manuscript to improve accessibility.

13- While the model shows high discrimination (ROC-AUC), no clinically meaningful cutoffs, thresholds, or risk bands are proposed. These are essential for translating predictive outputs into actionable decisions.

**Have the authors made all data and (if applicable) computational code underlying the findings in their manuscript fully available?**

Reviewer #1: Yes

Reviewer #2: **No: ** The authors did not provide a public repository to access all data and computational code.

Reviewer #3: Yes

PLOS authors have the option to publish the peer review history of their article (what does this mean? ). If published, this will include your full peer review and any attached files.

**Do you want your identity to be public for this peer review?** For information about this choice, including consent withdrawal, please see our Privacy Policy .

Reviewer #1: No

Reviewer #2: No

Reviewer #3: No

**Figure resubmission:**
---

## [Decision Letter · Decision Letter 1]

12 Nov 2025

Dear Professor Iwami,

We are pleased to inform you that your manuscript 'Prediction of graft loss in living donor liver transplantation during the early postoperative period' has been provisionally accepted for publication in PLOS Computational Biology.

Best regards,

Hao Hu, Ph.D

Academic Editor

PLOS Computational Biology

Jennifer Flegg

Section Editor

PLOS Computational Biology

Reviewer's Responses to Questions

**Comments to the Authors:**

Reviewer #2: All of my concerns have been addressed.

Reviewer #3: I appreciate the authors for addressing my concerns. This version can be accepted for publication.

**Have the authors made all data and (if applicable) computational code underlying the findings in their manuscript fully available?**

Reviewer #2: None

Reviewer #3: Yes

PLOS authors have the option to publish the peer review history of their article (what does this mean? ). If published, this will include your full peer review and any attached files.

**Do you want your identity to be public for this peer review?** For information about this choice, including consent withdrawal, please see our Privacy Policy .

Reviewer #2: No

Reviewer #3: **Yes: ** Mohammad Sadegh Taghizadeh

---

## [Editor Report · Acceptance letter]

PCOMPBIOL-D-25-00261R1

Prediction of graft loss in living donor liver transplantation during the early postoperative period

Dear Dr Iwami,

I am pleased to inform you that your manuscript has been formally accepted for publication in PLOS Computational Biology. Your manuscript is now with our production department and you will be notified of the publication date in due course.

With kind regards,

Anita Estes
